# Extract Methods, Molecular Characteristics, and Bioactivities of Polysaccharide from Alfalfa (*Medicago sativa* L.)

**DOI:** 10.3390/nu11051181

**Published:** 2019-05-27

**Authors:** Chen Zhang, Zemin Li, Chong-Yu Zhang, Mengmeng Li, Yunkyoung Lee, Gui-Guo Zhang

**Affiliations:** 1Department of Animal Nutrition, Shandong Provincial Key Laboratory of Animal Biotechnology and Disease Control and Prevention, Shandong Agricultural University, 61 Daizong Street, Taian City 271018, China; zhang3602@hotmail.com (C.Z.); sdgrass@163.com (Z.L.); sdforage@163.com (C.-Y.Z.); 2Department of Dairy Science, Virginia Tech, Blacksburg, VA 24060, USA; mengli@vt.edu; 3Department of Food Science and Nutrition, Jeju National University, Jeju 63243, Korea; lyk1230@jejunu.ac.kr; 4Department of Nutrition, University of California Davis, One Shields Ave, Davis, CA 95616, USA

**Keywords:** polysaccharide, compositions, glycosidic linkage, molecular weight, immunomodulation

## Abstract

The polysaccharide isolated from alfalfa was considered to be a kind of macromolecule with some biological activities; however, its molecular structure and effects on immune cells are still unclear. The objectives of this study were to explore the extraction and purifying methods of alfalfa (*Medicago sativa* L.) polysaccharide (APS) and decipher its composition and molecular characteristics, as well as its activation to lymphocytes. The crude polysaccharides isolated from alfalfa by water extraction and alcohol precipitation methods were purified by semipermeable membrane dialysis. Five batches of alfalfa samples were obtained from five farms (one composite sample per farm) and three replicates were conducted for each sample in determination. The results from ion chromatography (IC) analysis showed that the APS was composed of fucose, arabinose, galactose, glucose, xylose, mannose, galactose, galacturonic acid (GalA), and glucuronic acid (GlcA) with a molar ratio of 2.6:8.0:4.7:21.3:3.2:1.0:74.2:14.9. The weight-average molecular weight (Mw), number-average molecular weight (Mn), and Z-average molecular weight (Mz) of APS were calculated to be 3.30 × 10^6^, 4.06 × 10^5^, and 1.43 × 10^8^ g/mol, respectively, according to the analysis by gel permeation chromatography-refractive index-multiangle laser light scattering (GPC-RI-MALS). The findings of electron ionization mass spectrometry (EI-MS) suggest that APS consists of seven linkage residues, namely 1,5-Araf, galactose (T-D-Glc), glucose (T-D-Gal), 1,4-Gal-Ac, 1,4-Glc, 1,6-Gal, and 1,3,4-GalA, with molar proportions of 10.30%, 4.02%, 10.28%, 52.29%, 17.02%, 3.52%, and 2.57%, respectively. Additionally, APS markedly increased B-cell proliferation and IgM secretion in a dose- and time-dependent manner but not the proliferation and cytokine (IL-2, -4, and IFN-γ) expression of T cells. Taken together, the present results suggest that APS are macromolecular polymers with a molar mass (indicated by Mw) of 3.3 × 10^6^ g/mol and may be a potential candidate as an immunopotentiating pharmaceutical agent or functional food.

## 1. Introduction

Alfalfa (*Medicago sativa* L.) is an extensively cultured, dual purpose plant that has been generally used as forage for livestock or a kind of functional food for humans due to its high-quality protein [1,2,3] and immunomodulation bioactivity [4,5]. Prior studies have reported that there are noncellulosic polysaccharides in alfalfa, and subsequent research also clarified that those polysaccharides exert immunomodulatory [6], anti-inflammatory [7,8], antioxidant/anticancer [9,10], and growth-promoting bioactivities [11,12]. In addition, some dietary fiber and indigestible phytogenic polysaccharides can increase intestinal microbiota biodiversity and reduce the incidence of chronic disease [13,14]. Also, the polysaccharides in alfalfa (APS) have been regarded as a natural alternative to antibiotics when added to animal diets because they promote growth performance and have no toxic or other side effects in animals [3,15,16]; also, they have been used as functional agents or foods [16,17]. In recent years, a number of studies concentrated on investigating the extraction method, molecular structure, and biological activities of APS [8]. Aspinall and Gestetner (1968) firstly documented that there are two kinds of polysaccharides in alfalfa involving the hemicellulose and pectin. Chen and Liu (2015) identified polysaccharides from alfalfa by cold alkali soaking as rhamnogalacturonan I (RG-I) type pectin, with a molecular weight of 2.38 × 10^3^ kDa and a radius of 123 nm, that were composed of four kinds of monosaccharides and exhibited anti-inflammatory properties. However, Rovkina and Krivoshchekov (2018) isolated the polysaccharide from alfalfa using aqueous extraction method and found this polysaccharide consists of three monomeric compositions with molecular weights of 1100 ± 60, 16 ± 2, and 7 ± 1 kDa. Similarly, Li, and Tang (2013) reported that the bioactivation of polysaccharide from alfalfa using aqueous extraction was to enhance the energy metabolism and proliferation of mouse bone marrow dendritic cells and peritoneal macrophages of mouse. Thus, the extract procedure might impact the compositions and biological activation of APS. 

In addition, studies on other plant-derived polysaccharides also found the components and structural characterization vary due to different extraction methods [7] and plant cultivation conditions [15,18]. Meanwhile, the biological functions of polysaccharides are intimately correlated with their composition and molecular characteristics [13,19]. Thus, it is vital to explore the appropriate extraction procedure, molecular properties and biological activities of APS for the amplifying the application as function food or agent. Additionally, in cell cultures, high-purity polysaccharides are necessary to identify the specific biological function of the polysaccharides rather than the other components. Therefore, it is important to build a feasible method for extracting and purifying alfalfa polysaccharides (APS) to develop new functional foods or pharmaceutical agents [13,19]. Recently, there has been an increasing number of studies that have focused on the extracting technologies, biological activities [20,21,22], and molecular mechanisms of various plant-derived polysaccharides [8,23]. However, there is still some controversy surrounding the relationship between the components and bioactivities of APS. In our previous study, it was ascertained that the antioxidant properties of APS which could relieve the oxidant stress in mouse embryonic fibroblasts (MEFs) induced using H_2_O_2_ by activating the MAPK/P38 pathways and inhibiting the NK-κB pathways [8,23]. Meanwhile, a number of plant-derived polysaccharides have been suggested to modulate immune system function and have been applied in animals or humans as an immune-enhanced adjuvant [24]. A few studies have reported that dietary polysaccharides can change the gut microbiota and immunity of animals [7,25]. However, most studies have focused on the effect of oral or injected polysaccharides on an animal’s immunity, while their impact on immune cells in vitro remains unclear. Furthermore, findings related to oral polysaccharides have been highly heterogeneous and insufficient to support broad product structure/function generalizations [7,26].

Therefore, it is vital to investigate the molecular structure and biological activity of APS, to provide support for expanding their application as a functional food or pharmaceutical agent. The objectives of this study were to explore an extraction and purifying technology to isolate polysaccharides from alfalfa and to decipher their molecular characteristics, as well as to ascertain their bioactivities related to the proliferation and immunity functions of mouse splenic B and T lymphocytes in vitro.

## 2. Materials and Methods

### 2.1. The Extraction and Purification of Polysaccharides from Alfalfa

Alfalfa polysaccharide (APS) was prepared using the methods described by Park and Shin (2016) and Zhang and Gan (2019) with substantial modifications to simplify the procedure and to enhance the yield of polysaccharides (Chinese national invention patent, Patent No. 201610415782.4). Briefly, fresh alfalfa samples were chopped to ~5 cm in length and dried in an air-forced oven at 65°C. The dried sample was then mixed with double-distilled water in the ratio of 1:10 (alfalfa:distilled water), boiled for 1 hour, and subsequently filtered through two layers of nylon mesh (0.2-cm mesh). The liquid fraction was centrifuged at 1000× *g* for 10 min, and the supernatant was transferred to another container and added to 4 volumes of absolute ethyl alcohol (*v*/*v*). The mixture was kept at 4 °C for 12 hand then centrifuged at 3000× *g* for 10 min to precipitate crude polysaccharides. The crude polysaccharide pellet was subsequently redissolved in d-H_2_O and dialyzed using a biological semipermeable membrane (8000-M_W_ cutoff, Beijing Solarbio Science and Technology Co., Ltd., Beijing, China) against d-H_2_O (10 times the sample volume) at 4 °C for 2 days, with changing the d-H_2_O every 12 h. After dialysis, the extract remained in the dialysis bag was further deproteinized twice with chloroform:butanol (5:1, *v*/*v*) following the modified Sevag method [27,28,29]. The obtained material was precipitated with absolute ethyl ethanol, and centrifuged again as described above. The sediment was collected and lyophilized using a vacuum dryer (Biosafer-10A, Biosafer, Najing, China), which was considered as purified polysaccharide.

In total, five samples of alfalfa at the early blooming (10% blooming) stage from five different farms were obtained (one composite sample per farm) and used in this study. The extraction and the following composition and glycosidic linkage determination were performed three times and the mean value was used for each sample. The sample was treated as experimental unit. The detailed procedure of extracting and purifying APS is shown in Figure 1. The resulting product was a hazel colored powder containing 96.38% (*wt*/*wt*) polysaccharide, which was analyzed using galactose as the standard by the anthrone–sulfuric acid method [27,30].

### 2.2. Determination of Monosaccharide Components of APS

All polysaccharides are formed through a series of monosaccharides or uronic acid polymerized by glycosidic linkages, which can be broken by strong acid acidolysis; the components of polysaccharides can be identified by ion chromatography (IC).

Firstly, the absorption curve was built by determining the absorption of the standard substrate of monosaccharides, allowing for fucose, arabinose, galactose, glucose, xylase, mannose, fructose, ribnose, galacturonic acid (GalA), and glucuronic acid (GlcA) in this study. The standard sample of each monosaccharide was accurately weighed as 10.00 mg, dissolved in 10 mL of ultrapure water in 10-mL volumetric flasks, and dissolved in d-H_2_O to get 1, 5, 8, 16, 30, 40, 50, and 60 μg/mL standard solutions. Each standard solution was analyzed by IC (Thermo Fisher Scientific, ICS5000 type, Waltham, MA, USA) and a diode-array detector (DAD; Young-Lin Co. Changsha, China). The flowing phases were NaOH (200 mM) or NaAC (200 mM) and the flowing rate was 1.0 mL/min. The sample injection volume was 100 μL. The standard curve for each monosaccharide was formed according to the retention time and absorption peak.

A 10-mg (±0.005) APS sample was accurately weighted into a glass tube, to which 1 mL of trifluoroacetic acid (TFA) was added for 90-min acidolysis at 110 °C. The acidolysis mixture was dried in a vacuum rotator, and 5 mL of sterile water was added to fully dissolve the residue. The resulting solution was centrifuged at 12,000× *g* for 10 min, and the supernatant was collected to determine the components by IC following the above mentioned procedure. The retention time was matched with the monosaccharide standard curves to decipher the monosaccharide composition of APS. The molar amount of each component was calculated from the peak area of each derivative.

### 2.3. Determination of Molecular Weight of APS

Molecular weight is a vital factor which represents specific molecular, physical, and chemical characteristics of polysaccharides. Even some biological activities are also decided by molecular weight. It can be described according to the following parameters; the number-average molecular weight (Mn), expressed as Mn = Σ(niMi)/Σni, the weight-average molecular weight (Mw) defined as Mw = Σ(niMi2)/Σ(niMi), and the *Z*-average molecular weight (Mz), calculated as Mz = Σ(niMi3)/(ΣniMi2) [31]. The polydispersity coefficient presents the range of the molecular mass distribution and can be calculated by dividing Mw by Mn or dividing M_Z_ by Mn. The purified APS samples were accurately weighed as 5.00 mg and dissolved in 1 mL of 90% DMSO in a glass tube (HPLC, 543900, Merck, Waltham, MA, USA) for an overnight incubation in a 100 °C water bath, to which 3 mL of absolute ethanol (analysis of pure, Sinopharm, Beijing, China) was added and then mixed vigorously. The mixture was centrifuged at 1000× *g* for 5 min and the supernatant was removed. Then, the residual sediment was rinsed twice with anhydrous ethanol and 3 mL of 0.1 M NaNO_3_ (containing 0.02% NaN3221341, Sigma Aldrich, Shanhai, China) was added for a 20 min incubation at 121 °C. The mixture was centrifuged at 12,000× *g* for 10 min, and the supernatant was collected and determined by the gel permeation chromatography-refractive index-multiangle laser light scattering (GPC-RI-MALS) (DAWN HELEOS II, Wyatt Technology, Santa Barbara, CA, USA) method. The eluent was a 0.1 M NaNO_3_ and 0.02% NaN_3_ mixture (1:1, *v*/*v*) (HPLC, 221341, Sigma-Aldrich, Shanhai, China). The flow rate was maintained at 0.4 mL/min, and the temperature of the column was maintained at 60 °C. The eluent was monitored with a refractive index detector (RI, RID Agilent 1260); the analytic column included an Ohpak SB-805 HQ, Ohpak SB-804 HQ, Ohpak SB-803 HQ (Shodex, Asahipak, Tokyo, Japan). The sample injection volume was 300 µL. Mass spectra were viewed and analyzed by ASTRA6.1 software (Wyatt Technology Corporation, Santa Barbara, CA, USA).

### 2.4. Determination of Glycosidic Linkages of APS

The APS samples were firstly treated by partially methylated alditol acetates (PMAAs) following a previously reported protocol [8,32]. In this procedure, free hydroxyl groups were completely methylated on the glycan with the glycosidic linkages remaining intact. The resulting methylated glycan was further hydrolyzed into monosaccharides, which were then converted to PMAAs by reduction and acetylation. The PMAAs were analyzed by electron ionization mass spectrometry (EI-MS) to identify glycosidic linkages between the adjacent monomers.

Following the EI-MS method, polysaccharides were firstly treated through methylation, acetylation, hydrogenation, deuterium, and other processes to transform them into volatile derivatives, and then the long chains of derivatives were broken into fragments by electron bombardment. By analyzing the fragments, the position of the free hydroxyl in the derivatives was determined, and the linkage mode of each monosaccharide residue was finally defined. Glycosidic linkages of APS were identified by comparing the mass spectrum patterns of PMAAs with the database of CCRC.32. The molar ratios of individual linkage residues were calculated based on peak areas and response factors for each sugar in the Total Ion Chromatography (TIC), as described previously [8,32]. The EI-MS spectrum of APS is shown in Figure 2.

### 2.5. The Immunomodulation to Mouse Splenic T and B Lymphocytes

The immunomodulation properties of APS were evaluated by determining their effects on proliferation and cytokine expression in cultured T and B lymphocytes [33]. Briefly, the spleen lymphocytes were isolated from RAW 264.7 mouse, which were free of red blood cells from treatment with a lysis buffer (0.15 M NH_4_Cl, 0.01 M KHCO_3_, and 0.1 mM Na_2_EDTA, pH 7.4). To remove adherent cells, such as macrophages, the spleen cells were incubated for 1 hour in Petri dishes at 5 × 10^6^ cells/mL. The B and T cells were separated using the nylon wool method (Polysciences Inc., Warrington, PA, USA) according to the manufacturer’s instructions [34]. After washing with PBS buffer solution and counting with a counter board, the isolated B and T lymphocytes were used for subsequent assays.

#### 2.5.1. Proliferation Assay of Mouse Splenic T and B Cells

The separated T and B cells were incubated in plates with at least three replicates to determine the effects of APS on proliferation. Complete RPMI-1640 medium containing 10% FBS and 1% antibiotic was used as a vehicle control. Briefly, the freshly isolated T or B cells (2 × 10^5^ cells/mL) were cultured in flat-bottom 96-well plates with a volume of 100 μL per well with complete RPMI-1640 medium and treated with a series of final concentrations (0–30 μg/mL) of APS or the specific lymphocyte mitogens Congestin A (Con A)/Lipopolysaccharide (LPS) (specific T/B cell mitogen, 10 μg/mL, Sigma Chemical Co., St. Louis, MO, USA), respectively. Cells were incubated at 37 °C and 5% CO_2_ for 12, 24, and 48 h. Then, 10 μL of MTT (Sigma, USA, 5 mg/mL) was added to each well and incubated for another 4 h in the dark. The formazan crystals presented in the cells were dissolved in 50% dimethyl sulfoxide (Solarbio, Beijing, China) solution (pH 7.4) containing 10% sodium dodecyl sulfate (*w*/*v*). After the plates were shaken at 480 × *g* for 10 min, the absorbance was measured at 490 nm in a microplate reader (Thermo Multiskan MK3, Thermo Fisher Scientific Inc., Waltham, USA). All determinations were performed for six replicates for every sample, and three independent assays were conducted.

#### 2.5.2. IgM Production of B Cells

The separated B cells were incubated in 24-well plates (1 mL per well) with APS at final concentrations of 0 to 30 µg/mL at three different incubation times of 12, 24, and 48 h. Then, a total of 1.0 mL of 0.2% SRBCs and 1 mL of diluted guinea pig complement (1:20 in PBS, Bianzhen Biological Technology Co., Ltd., Nanjing, Jiangsu, China) were added to each well and incubated for 1 hour at 37 °C. The cell suspensions were then collected and centrifuged at 1000× *g* for 5 min, and the optical density of the supernatants was determined at 405 nm using a microplate reader (MK3, Thermo). A blank control supplemented with 1 mL of complete RPMI-1640 medium in place of B cell suspensions was used in each determination. All determinations were performed using six replicates per sample. The production of IgM was expressed using the optical density (OD) value.

#### 2.5.3. The mRNA Expression of IL-2, IL-4, and IFN-γ in T Cells

Cytokine expression levels were used to determine T cell activation. T cells (5 × 10^6^ cells/mL) were cultured in 12-well plates and incubated with APS (5, 10, 20, or 30 µg/mL) or Con A (10 µg/mL) for 24 h. Con A treatment was used as a positive control. Total RNA was extracted using an Ultraspec II RNA Isolation Kit (Biotech Lab., Houston, TX, USA). Reverse transcription polymerase chain reaction (RT-PCR) was performed to determine changes in cytokine gene expression as described previously [34]. The RNA was reverse transcribed using a GeneAmp RNA PCR kit using 100 ng of total cellular RNA (Perkin-Elmer, Branchburg, NJ, USA). PCR was carried out using 2.5 units of AmpliTaq DNA polymerase in a Bio-Rad Cycler (Bio-Rad Lab., Richmond, CA, USA). PCR products were electrophoresed on a 3% Nusieve 3:1 agarose gel and photographed after staining with ethidium bromide.

### 2.6. Statistical Analysis

The normality of observed data was verified using the Shapiro–Wilk test, and all the data were normally distributed. Results were expressed as the mean ± standard deviation (SD) with three replicates per sample. The significance of differences between means were evaluated by one-way analysis of variance (ANOVA) and Tukey’s multiple comparison test using SAS version 9.0 (SAS Inst. Inc., Cary, NC, USA). Statistical significance was declared at *p* < 0.05.

## 3. Results

### 3.1. The Monosaccharide Compositions of APS

The yield ratio of purified polysaccharides extracted from alfalfa was about 15.76% (*w*/*w*) dry weight of whole-plant alfalfa, and the content of total sugar was 96.38% and protein was 3.62% (Table 1). The compositions of APS were determined using IC and by matching the retention time in the chromatographic curve with that of the monosaccharide standard substance. Eight peaks were identified in APS in the order of fucose, arabinose, galactose, glucose, xylose, mannose, galactose, GalA, and GlcA with a molar ratio of 2.6:8.0:4.7:21.3:3.2:1.0:74.2:14.9 (Figure 3, Table 1). Of note, that the molar proportions of fucose, arabinose, and mannose (denoted with dotted lines) were less than 3%, and those compositions might vary considerably in different samples or extracting methods. Arabinose, galactose, glucose, GalA, and GlcA (denoted with solid lines) were the main components.

### 3.2. Molecular Weight

A polymer, including the polysaccharide, does not have a fixed molecular weight but a mixed system of homologs with different molecular weights. Therefore, the molecular weight of a polymer is distributed in a range and generally expressed as the average value. This heterogeneity of molecular weight is called the polydispersity of the polymer, which is usually characterized by multiple dispersion coefficients, which is the ratio of the weight-average molecular weight to the number-average molecular weight, or the ratio of the *Z*-average molecular weight to the weight-average molecular weight.

The molecular weights of APS are presented in Table 2, following analysis by GPC-RI-MALS. The Mw, Mn, and Mz of APS were calculated to be 3.30 × 10^6^, 4.06 × 10^5^, and 1.43 × 10^8^ g/mol, respectively. The polydispersity coefficient was 8.14 (Mw/Mn) or 353.31 (Mz/Mn). In addition, the radius mean square of APS was simultaneously determined to be 53.4, 51.8, and 59.0 for the number-average radius (Rn), weight-average radius (Rw), and Z-average radius (Rz), respectively.

Figure 4 shows the variation trend of molar mass, laser scattering (LS), and refractive index (RI) with the retention time of APS. The rapidly decreasing LS curve (red line) and gradually increasing RI line (blue line) indicate that the polysaccharides contained fewer proportional macromolecule polysaccharides and a much greater proportion of small-molecule polysaccharides (the molar mass has a polydispersity). This was consistent with the changing curve of the molar mass line (black line), which decreased gradually with the extension of the retention time. The intersection of LS and RI curves corresponded with the value of the molar mass line of ~10^6^, which reflected the distribution range of the main molecular weight.

### 3.3. Glycosidic Linkages between Monosaccharides

The mass spectrum analysis using EI-MS suggested that APS was mainly composed of seven different linkage residues, including nonreducing terminals of galactose (T-D-Glc) and glucose (T-D-Gal), the molar ratios of which were 4.02% and 10.28%, respectively (Table 3). Other saccharide residues, including 1, 5-Araf; 1, 4-GalA; 1, 4-Glc; 1,6-Gal; and 1,3,4-GalA with molar ratios of 10.30%, 52.29%, 17.02%, 3.52%, and 2.57%, respectively, were also identified as the intrachain residues. In addition, the degree of branching (DB) value was 16% for APS, which was the proportion of the accumulated number of the terminal and branch residues accounting for the total amount of saccharide residues. The terminal residues of T-Glc and T-Gal were linked at the branch or the end of the main chain by 1,4-glycosidic bonds. The proposed repeating units of APS and molecular structure are shown in Figure 5.

### 3.4. APS Increased Proliferation and Immune Functions of B but Not T Cells

The immunomodulatory bioactivity of APS was evaluated by determining their effects on the proliferation and cytokine expression of splenic T and B lymphocyte cells stimulated with APS. The proliferation of B cells treated with APS at different concentrations (0, 5, 10, 20, or 30 µg APS/mL) and different incubation times (12, 24, or 48 h) is presented in Table 4. Within the same incubation time, adding 10 µg/mL of APS indicated greater (*p* < 0.05) proliferation than that of the other concentrations but less than the LPS-treated group. On the other hand, regardless of APS content, the APS-treated B cells presented greater proliferations in 24 h of incubation than those of 12 and 48 h of incubation. In addition, the IgM concentration in cultured B cells for 24 h at 10 µg/mL of APS was similar to that of the LPS group and higher that of other groups (*p* < 0.05). The effects of APS on T cells were then investigated to confirm that the activation of APS to T cells. The results showed (see Table 4) that ASP did not activate T cells. Splenic lymphocytes were incubated with APS for 24 h and the expressions of IL-2, IL-4, and IFN-γ mRNAs were determined. As shown in Figure 6, the mRNA expressions of IL-2 and IL-4 were unaltered by the APS treatment, whereas Con A, used as a reference T cell activator, increased these gene expressions. In summary, these results suggest that APS selectively activates B cells in a dose- and time-dependent manner, but does not affect T cells.

## 4. Discussion

In the present study, APS was fractioned and purified using a new modified method, and it consisted of six monosaccharides and two uronic acids, namely fucose, arabinose, galactose, glucose, xylose, mannose, galactose, GalA, and GlcA; the molar ratio was 2.6:8.0:4.7:21.3:3.2:1.0:74.2:14.9 and the average Mw was 3.3 × 10^6^ g/mol. The findings had some inconsistencies in terms of composition and proportion with previous studies. isolated a kind of polysaccharide from alfalfa using a cold alkali method, and it was identified as an RG-I type pectin with a molecular weight of 2.38 × 10^6^ g/mol. This kind of polysaccharide consisted of four monosaccharides of rhamnose, galactose, galactouronic acid, and arabinose in a molar ratio of 2.4:1.4:1.0:3.4 and exerted novel anti-inflammatory bioactivity. Whereas Rovkina and Krivoshchekov (2018) observed that the polysaccharide from alfalfa using an aqueous extraction method was composed of three monomeric compositions, namely arabinose, glucose, and galacturonic acid with molecular weight of 5.0 × 10^6^ g/mol. In addition, the intrachain glycosidic linkages of APS obtained in this study contained (1→3, 4), (1→6), and (1→5) glycosidic bonds between the adjacent monosaccharide, and GalA residue was the main component (with a ratio of 51.29%) in the chain. This was in agreement with previous reports on the molecular structural characterization of alfalfa polysaccharides indicating the largest proportions of D-galacturonic acid based on partial acid hydrolysis and/or partial acetolysis analysis [1,8,35]. Thus, it can be seen that these three kinds of polysaccharides from the same plant are different in composition while having the same order of magnitude of 10^6^ in Mw, suggesting a similar macromolecular and the same main intrachain component of galacturonic acid residue accounting for more than 50%. The discrepancy in composition might be attributed to the different extraction methods (boiling water [1] vs. cool alkali [8]) that isolated polysaccharide molecules of different sizes and varieties. 

The molecular characteristics and compositions of polysaccharides are affected by the plant species and the extraction method [20,36]. For example, the polysaccharides isolated from the mushroom *Dictyophora indusiata* with boiling water were composed of nine components, and the largest proportional monosaccharide residue was glucose (59.84%) [24]. Other polysaccharides derived from *Russula alatoreticula* (a kind of mushroom) with alkali treatment consisted of glucose, galactose, and mannose [24,25]. In addition, Park and Shin (2016) observed that the polysaccharides extracted from hallabong (*Citrus sphaerocarpa*) peels with water extraction and pectinase consisted of eight monosaccharides, and the molecular weight was at 10^3^ orders of magnitude. Lastly, Zheng and Zhang (2016) documented that two types polysaccharide (P2 and P3) were obtained from *M. kwangsiensis* Figlar & Noot by hot water extraction, and the main components were glucose for P2 and xylose and rhamnose for P3, as well as there being α-pyran ring or pyran group in the molecular structure. Thus, the extract methods and plant varieties are key factors in determining the compositions and characteristics of polysaccharides. In addition, the present findings indicated that the composition, glycosidic linkage, and molecular weight of APS had no significant difference among alfalfa samples from different area with the same development period (early blooming period). Rovkina and Krivoshchekov (2018) indicated the composition and molecular characteristics of APS did not differ significantly between years. Further studies should be conducted to investigate the variation of composition and bioactivities of APS between the different growth periods. 

The molecular characteristics and structure of polysaccharides are diverse due to their different compositions, linkage types, and molecular weights, which together have substantial effects on bioactivity [13,23]. In this study, the intrachain glycosidic linkages contained some (1→3, 4), (1→5), and (1→6) bonds, which have some similarities with the previously reported APS bonding structure; that is, the linkage between adjacent monosaccharide residue is not only a 1,4 glycosidic linkage. This kind of molecular linkage is different from starch (merely containing 1,4 glycosidic linkage) and makes these polysaccharides incapable of being digested by gastrointestinal enzymes in animal digestive tract. However, they can be partly fermented by specific intestinal microbiota to generate short-chain fatty acids (SCFAs). Short-chain fatty acids have been documented to be beneficial for enhancing gut barrier integrity and immunity regulation [37]. Additionally, in the present study, APS selectively increased the proliferation of B cells, but not T cells, in a dose- and time-dependent manner and also enhanced IgM cytokine secretion. Thus, similar to LPS, APS specifically activated the B cell as a mitogen. The time- and dose-dependent biological activation of APS was consistent with that of other plant-derived bioactive compounds, such as mistletoe lectin (KML-C) [28] and polysaccharides from Russula [25] and Acanthopanax senticosus [34]. The molecular mechanisms by which APS active B cells need further explorations to elucidate the corresponding membrane receptors and intracellular signaling pathways. 

Certain polysaccharides generally exhibit multiple biological activities [38]. Chen and Liu (2015) found that polysaccharides isolated from alfalfa by cold alkali had anti-inflammatory property, while a kind of water-soluble polysaccharide derived from alfalfa increased the total lymphocytes proliferation [36]. Likewise, in a parallel study, we have demonstrated an antioxidant activation of APS in attenuating the oxidant stress of MEFs cells by modulating the MAPK/p38 and NF-κB pathways [9]. Similarly, the polysaccharides isolated from the mushroom *D. indusiata* presented protective effects for the restoration of antibiotic-driven gut microbiota dysbiosis and enhancement of gut barrier integrity [25]. Polysaccharides derived from Korean Citrus hallabong peels could block tube formation of human umbilical vein vascular endothelial cells and inhibited breast cancer cell migration through downregulation of matrix metalloproteinase 9 in MDA-MB-231 triple-negative breast cancer cells [27]. Thus, polysaccharides derived from different plants exhibit diverse biological activities based on their different molecular composition and structure [7]. Based on the current findings, unlike the 1,4- glycosidic linkage of starch, the molecules of APS have a special linkage and molecular structure, which means APS cannot be digested by enzymes in an animal’s gut. This may be the crucial molecular basis for its biological activities, and it has been documented in our prior experiment that APS can bind to the membrane receptor of toll like receptor4 in MEF cells and modulate some gene expression by activating relative signaling pathways [9].

The current results indicate that APS can directly activate B cell proliferation and enhance cytokine secretion; although it remains unclear which signaling pathways are activated in this process. It is also unknown whether, in vivo, the small molecules produced by intestinal microorganisms that degrade APS are also effective ways by which APS regulates animal immunity. Thus, future research is worthy of investigating the molecular mechanisms of immunomodulation and potential biological activities of APS, as well as the interaction of APS with gut microbiota and its effects on metabolome.

## 5. Conclusions

In conclusion, a high-purity macromolecular polysaccharide with the average Mw of 3.3 × 10^6^ g/mol, containing two uronic acids and seven monosaccharides, can be obtained from alfalfa (APS) using the hot water extraction, ethanol precipitation, and dialysis methods. The intrachain glycosidic linkage within the APS molecule mainly consists of T-D-Glc, T-D-Gal, 1, 5-Araf, 1, 4-GalA, 1, 4-Glc, 1,6-Gal, and 1,3,4-GalA residues. 

APS enhanced the proliferation of B cells and the secretion of immune cytokines in a dose- and time-dependent manner. Thus, the findings of this study on APS can provide a scientific reference for applying natural polysaccharides in pharmaceutical and functional food.

## Figures and Tables

**Figure 1 nutrients-11-01181-f001:**
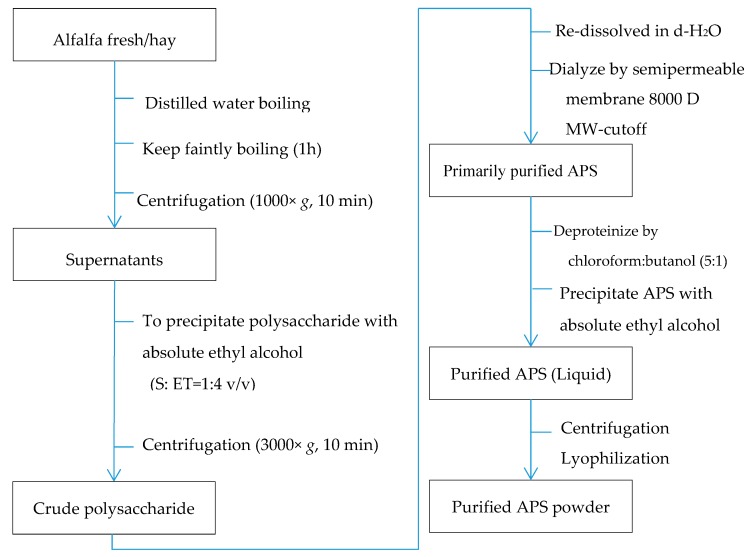
Extraction and purification procedure of alfalfa polysaccharide (APS).

**Figure 2 nutrients-11-01181-f002:**
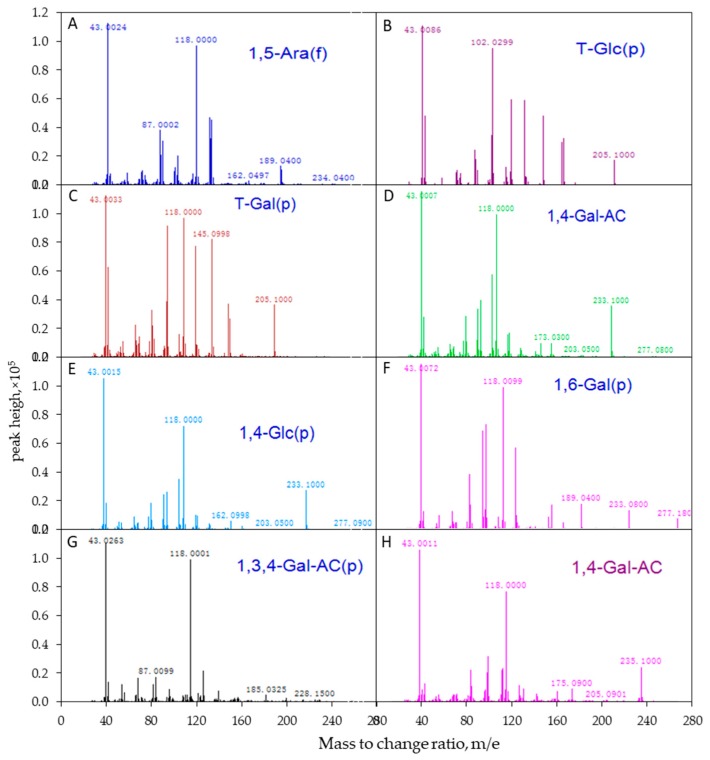
The mass spectrogram of each intrachain residue of APS molecular determined using electron ionization mass spectrometry (EI-MS). The polysaccharide samples were determined by EI-MS after hydrogenation (**A**–**G**)/treated with deuterium (**H**). (**A**) (1,5)-furan arabinose residue; (**B**) terminal glucose residue; (**C**) terminal galactose residue; (**D**) (1,4)-galacturonic acid; (**E**) (1,4)-glucose residue; (**F**) (1,6)-galactose residue; (**G**) (1,3,4)-galacturonic acid; (**H**) (1,4)-galacturonic acid (deuterium generation).

**Figure 3 nutrients-11-01181-f003:**
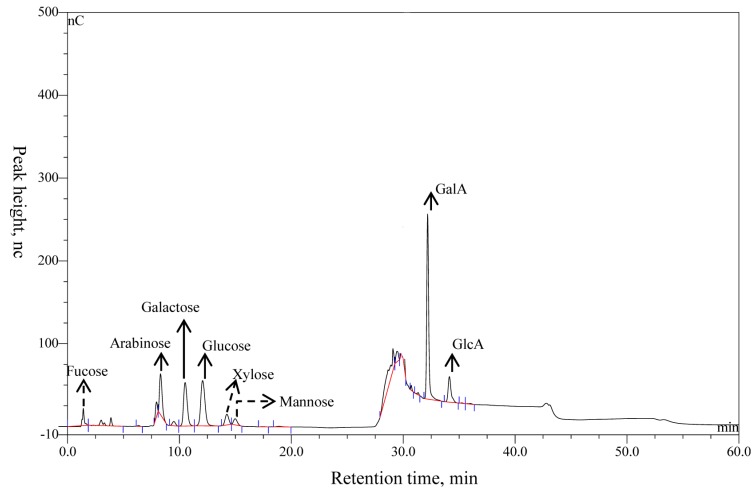
The ion chromatogram of APS.

**Figure 4 nutrients-11-01181-f004:**
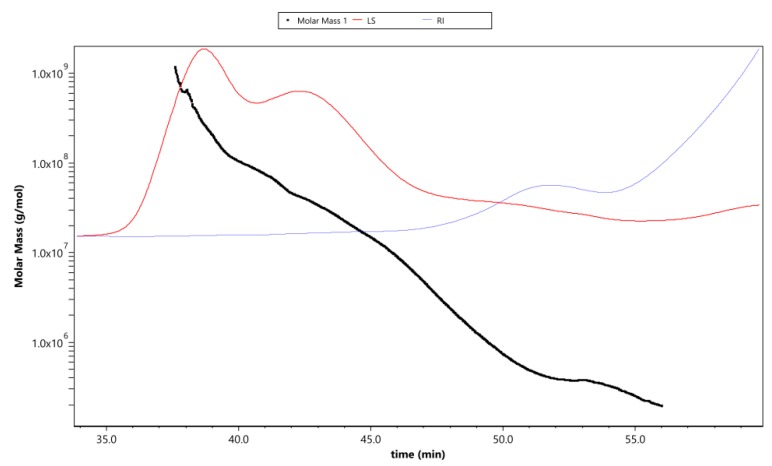
The variation tendencies of molar mass, laser scattering (LS), and refractive index (RI) of APS. The red line shows the varying tendency of the LS of APS with the retention time, and the blue line represents the trend of the RI of APS. The tendency of the red and blue lines indicates the size of the polysaccharide molecules and their relative proportions contained in the tested sample. The black line is the varying tendency of the molar mass of the polysaccharides following the retention time.

**Figure 5 nutrients-11-01181-f005:**
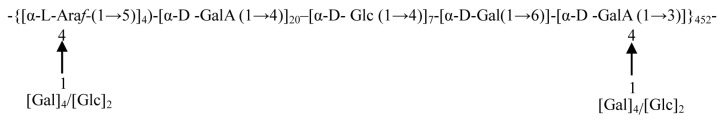
Proposed repeating units of APS and molecular structure.

**Figure 6 nutrients-11-01181-f006:**
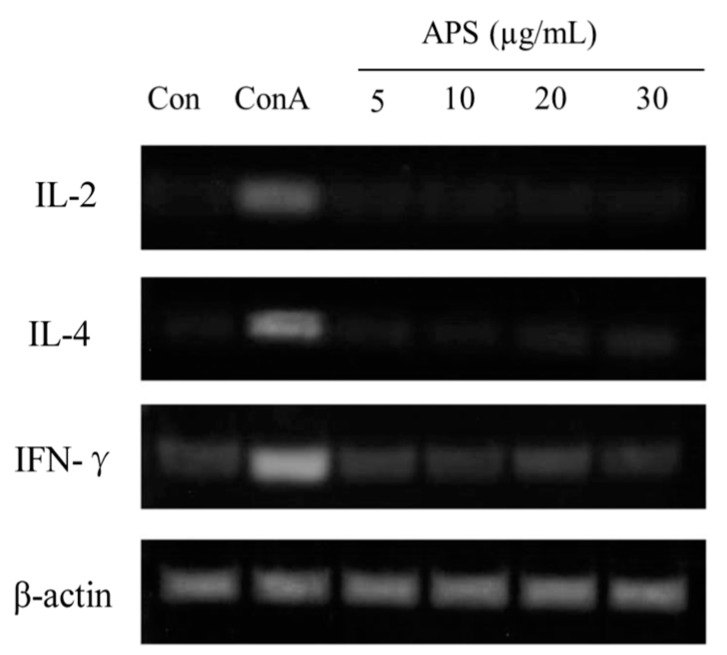
Cytokine expressions of IL-2, IL-4, and IFN-γ in cultured T cells for 24 h incubation with increasing APS concentrations.

**Table 1 nutrients-11-01181-t001:** The chemical components of polysaccharides fractionated from alfalfa ^1^.

Chemical Component	Neutral Sugar (%)		Protein (%)		Yield of Crude Polysaccharide (*w*/*w*) %	
	96.38 ± 4.35		3.62 ± 0.48		15.76 ± 1.67	
Component	RT, min	nc	nc × time	Relative Peak Area	μg/mg	mol/mol, %
Fucose	3.82 ± 0.08	9.84 ± 0.08	3.27 ± 0.19	1.75 ± 0.05	4.38 ± 0.06	2.02 ± 0.02
Arabinose	8.29 ± 0.02	49.45 ± 0.03	13.28 ± 0.04	6.79 ± 0.10	12.40 ± 0.12	6.25 ± 0.06
Galactose	10.51 ± 0.01	52.83 ± 0.21	20.27 ± 0.41	10.26 ± 0.20	8.72 ± 0.07	3.66 ± 0.02
Glucose	12.05 ± 0.08	54.68 ± 0.08	24.44 ± 0.35	12.35 ± 0.12	39.20 ± 0.09	16.46 ± 0.03
Xylose	14.22 ± 0.03	12.68 ± 0.37	4.45 ± 0.33	2.48 ± 0.21	4.69 ± 0.11	2.36 ± 0.05
Mannose	14.91 ± 0.11	7.58 ± 0.14	2.80 ± 0.06	1.68 ± 0.21	6.74 ± 0.16	0.77 ± 0.02
GalA	32.16 ± 0.06	223.51 ± 9.31	47.11 ± 0.92	23.75 ± 0.13	146.63 ± 0.18	57.13 ± 0.03
GlcA	34.17 ± 0.04	30.83 ± 0.12	9.27 ± 0.08	4.63 ± 0.20	29.16 ± 0.22	11.36 ± 0.08

^1^ RT: retention time, nc: nano coulomb (the unit of quantity of electric charge), nc×time: the peak area calculated by integration.

**Table 2 nutrients-11-01181-t002:** The molecular weight, polydispersity, and radius mean square of APS.

Item		Values
Molar mass (g/mol)	Mw	(3.301 ± 0.012) × 10^6^
	Mn	(4.057 ± 0.013) × 10^5^
	Mz	(1.433 ± 0.036) × 10^8^
Polydispersity	Mw/Mn	8.14 ± 0.02
	Mz/Mn	353.31 ±13.77
Radius mean square (R.M.S.), nm	Rn	53.4 ± 3.3
	Rw	51.8 ± 3.3
	Rz	59.0 ± 3.0

**Table 3 nutrients-11-01181-t003:** The glycosidic linkages among the saccharide residues and molar proportions^1^.

Serial Number	Retention Time	nc	nc × min	Glycosidic Linkages	mol %
1	11.64	9,757,235	2,036,177	1,5-Ara(f)	10.28
2	12.88	4,013,392	749,606	T-Glc(p)	4.02
3	13.55	10,293,423	2,283,312	T-Gal(p)	10.30
4	15.83	56,765,889	10,600,206	1,4-Gal(p)NAC	52.29
5	16.02	18,470,747	4,355,016	1,4-Glc(p)	17.02
6	17.51	3,820,551	722,873	1,6-Gal(p)	3.52
7	17.89	3,016,060	450,777	1,3,4-Gal(p)NAC	2.57

^1^ RT: retention time, nc: nano coulomb (the unit of quantity of electric charge), nc×time: the peak area calculated by integration.

**Table 4 nutrients-11-01181-t004:** Effects of APS on the proliferation of B and T cells, and the IgM concentration in cultured B cells (OD values at 490 nm) ^1^.

Incubation Time, h	CON	LPS/ConA	APS concentration, µg/ mL			
			5	10	20	30
**B cells**						
**12**	0.018b ± 0.002	0.033 ** ± 0.003	0.021 ± 0.001	0.027 * ± 0.002	0.024 ± 0.002	0.024 ± 0.001
24	0.026 ± 0.003	0.0558 ** ± 0.003	0.031 ± 0.002	0.037 * ± 0.003	0.029 ± 0.003	0.033 ± 0.003
48	0.012 ± 0.001	0.024 * ± 0.004	0.017 ± 0.002	0.023 * ± 0.002	0.019 ± 0.002	0.018 ± 0.002
T cells						
12	0.048 ± 0.0105	0.152 ** ± 0.015	0.061 ± 0.014	0.059 ± 0.008	0.056 ± 0.012	0.058 ± 0.009
24	0.081 ± 0.011	0.163 ** ± 0.019	0.085 ± 0.006	0.089 ± 0.004	0.09 ± 0.005	0.089 ± 0.006
48	0.028 ± 0.007	0.139 ** ± 0.024	0.031 ± 0.007	0.034 ± 0.007	0.032 ± 0.006	0.04 ± 0.008
IgM (cytokine of B cells)						
24	0.391 ± 0.014	0.4107 ± 0.012	0.391 ± 0.011	0.412 ± 0.007	0.401 ± 0.006	0.401 ± 0.010

^1^ Splenic B and T cells were stimulated with increasing concentrations of APS and different incubation times. The proliferation of B cells subjected to APS stimulation at 0 (control), 5, 10, 20, and 30 µg/mL doses for 12, 24, and 48 h of incubation; LPS was set as a positive control. The proliferation of T cells subjected to APS stimulation at 0 (Control), 5, 10, 20, and 30 µg/mL doses for 12, 24, and 48 h of incubations; Con A was set as a positive control. The IgM concentration in cultured B cells incubated for 24 h with increasing APS concentrations. The significance of data in the same row were presented with *, with *means *p* < 0.05, and **means *p* < 0.01.

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
