# Peer review of "Extract Methods, Molecular Characteristics, and Bioactivities of Polysaccharide from Alfalfa (Medicago sativa L.)"

_nutrients, 2019, doi:10.3390/nu11051181_

Round 1

Reviewer 1 Report

The manuscript entitled “Extract Methods, Molecular Characteristics, and Bioactivities of Polysaccharide from Alfalfa (Medicago sativa L.) ” presents interesting issue, but it requires some important corrections.

Major:

1.      Authors did not specify in their manuscript the number of batches and samples. It is hard to guess if the presented study is a “case study” description of only one sample or the material may be perceived as a representative. Without such information it is hard to conclude about the scientific value of the paper.

2.      For a number of analysis (all except for those presented in sub-chapters 2.5.1. and 2.5.2.) there is no number of repetitions indicated. As a result, it is hard to guess if the result was just random or may be perceived as a representative. Without such information it is hard to conclude about the scientific value of the paper.

General:

Some parts of the manuscript are shabbily prepared (e.g. lacking spaces, or redundant spaces, leadings, References section, etc.)

Authors should avoid personal forms (e.g. “we ascertained”) and they should use rather not personal ones (e.g. “it was ascertained”).

Abstract:

Authors should briefly justify their study indicating the background – such information should be added (one sentence will be enough).

Number of batches/ samples/ repetitions should be specified.

Introduction:

The sentences in the second part of the manuscript seem to be not “combined”. In a paragraph each sentence should “follow” the previous one, while in this part they seem as just random information.

It seems that Authors indicate that extracting procedure may influence structure and, as a result, also the biological activity. Authors should more precisely describe it.

In this section, Authors should justify the study, while presenting the history of the problem. In the present form, they present the history of their own past studies. Such justification is in appropriate. If their own studies are the only one that were conducted – it should be described.

Materials and Methods:

Number of batches/ samples/ repetitions should be specified.

Figure 1 – The photographs do not present any information – they should be removed from the scheme, as Authors should briefly present most important data with no illustrations.

If there is more than one sample/ repetition analysed (see above – it is not known), there are also following problems:

It seems, that Authors did not verify the normality of distribution for the assessed variables. Authors must verify the normality of distribution and specify the test applied for verification.

If the distribution is normal, the mean values should be presented (accompanied by SD), but if it is different than normal, the median, accompanied by minimum and maximum values should be presented – it should be specified that distribution is normal if it is. 

The applied test should be chosen taking into account the observed distribution.

Results:

The whole section should be corrected accordingly taking into account the major problems specified above.

If the distribution is normal, the mean values should be presented (accompanied by SD), but if it is different than normal, the median, accompanied by minimum and maximum values should be presented – it should be specified that distribution is normal if it is. 

The applied test should be chosen taking into account the observed distribution.

Figure 6 (a-c) – data should be rather presented as a table to be easier to follow.

Discussion:
The section should be corrected accordingly and Authors should objectively assess the value of their results taking into account the number of batches/ samples/ repetitions.

Authors should extensively discuss the limitations of their study.

Conclusions:

Authors should not reproduce results in this section, but they should present the broader conclusion instead (1-3 sentences will be enough).

Authors Contributions:

Authors should correct this section in order to indicate clearly which Author contributed. E.g. who is “GZ”? do Authors mean GGZ? Authors should be consistent.

Author Response

Dear editor and reviewers:

I would like to extend our thanks to you for taking the time to carefully review our manuscript (nutrients-486076) titled “Extract Methods, Molecular Characteristics, and Bioactivities of Polysaccharide from Alfalfa (Medicago sativa L.)”. We are grateful for the helpful feedback.

We have carefully read the comments and suggestions, and revised our manuscript line-by-line following those comments and the “Guidelines for Authors” of the Journal. Following reviewers’ comments, all corrected portions are highlighted with “Tracked Changes” pattern in the re-uploaded manuscript. Before submitting, we checked our article according to the ‘instructions to authors’ of the journal, and asked a professional English editor (MDPI English editing) to help us to revise it.

We hope that these revisions have improved the paper such that you now deem it worthy of publication in “Nutrients.

If you have any questions, please do not hesitate to contact us. Once again, thanks for your detailed comments and suggestions to our manuscript.

Sincerely,

Guiguo Zhang, Ph.D.

Department of Nutrition, University of California (UC-Davis),

One Shields Ave, Davis, CA 95616, USA;

Department of Animal Nutrition, Shandong Agricultural University

Taian City, Shandong Province, 271000, China  

Tel : +86-13954859206

Email: guzhang@ucdavis.edu ;   zhanggg@sdau.edu.cn;  

Reviewer 1

Comments and Suggestions for Authors

The manuscript entitled “Extract Methods, Molecular Characteristics, and Bioactivities of Polysaccharide from Alfalfa (Medicago sativa L.)” presents interesting issue, but it requires some important corrections.

Major:

1. Authors did not specify in their manuscript the number of batches and samples. It is hard to guess if the presented study is a “case study” description of only one sample or the material may be perceived as a representative. Without such information it is hard to conclude about the scientific value of the paper.

2. For a number of analysis (all except for those presented in sub-chapters 2.5.1. and 2.5.2.) there is no number of repetitions indicated. As a result, it is hard to guess if the result was just random or may be perceived as a representative. Without such information it is hard to conclude about the scientific value of the paper.

R: Thanks for your constructive comments.

The procedures described in the manuscript concerning polysaccharide extracting, determination of component, molecular weight, and glycosidic linkage would provide a detailed profiles and parameters to the readers on the determination process of alfalfa polysaccharides. In fact, in the process of sample determination, at least three samples were determined, for ensuring a good replication.

In general, the deciphered results of chemical compositions, molecular weight, and glycosidic linkage under the standardized determination procedure and parameter setting were relatively stable (i.e., retention time, peak area, glycolysis linking etc.). For example, the glycolysis linkage existed objectively and didn’t change due to the variation of measuring method and numbers. In addition, similar results to ours can be found in previously published papers (Chen et al., 2015; Park et al., 2016; Zheng et al., 2016; Kanwal et al., 2018), in which only a representative plot (i.e. chromatogram) and corresponding typical traits data (table) were presented to describe the molecular characteristics of targeted material.

In the present study, five batches alfalfa from 5 different areas were collected and utilized in extracting and subsequent compositions determination trial. In Table 2, the number of samples was listed (N=5) in the initial submitted manuscript.  

According to the reviewer’s instructions, we have re-checked the normality of observed data using Shapiro-Wilk test, and confirmed that all the data were normally distributed. Results were expressed as mean ± SD with 5 replicated determinations. The significance of differences between means was evaluated by one-way analysis of variance (ANOVA) and Tukey’s multiple comparison test using SAS version 9.0. We revised the description of the statistical method.

However, it is needed to be pointed out that the glycosidic linkages between components of APS were determined using electron ionization mass spectrometry (EI-MS) in the present study. Following the normal determining procedure, the APS samples were firstly subjected to hydrogenation (Figure 2A-G) and deuterium (Figure 2H) treatment; Regardless of the retention time, the saccharide residues in APS molecule are all same including galactose (T-D-Glc), glucose (T-D-Gal), 1, 5-Araf; 1, 4-Gal-Ac; 1, 4-Glc; 1,6-Gal; and 1,3,4-Gal-AC. The glycolysis linkages inter-residues exist objectively, under the normal determining parameters setting, the deciphered results retain constant. The retention time has no practical significance, only indicates the parameters of this present measurement. Under the same retention time, the finding presented the same glycosidic residue in 5 samples determination. The saccharide residue and glycosidic linkage remained unchanged. So in statement of results, only one representative result is listed. In addition, according to the other reviewer’s suggestion, the ‘nc’ and ‘nc*min’ values were rounded to the nearest whole number.

I hope the detailed explanation and revision will meet the requirements of reviewer. Once again, thanks for your constructive comments.

General:

Some parts of the manuscript are shabbily prepared (e.g. lacking spaces, or redundant spaces, leadings, References section, etc.)

Authors should avoid personal forms (e.g. “we ascertained”) and they should use rather not personal ones (e.g. “it was ascertained”).

R: Thanks for your insightful instructions.

We have double checked the manuscript and revised some mistakes, meanwhile, asked the MDPI English editing service to correct the style and language according the requirement of journal of “Nutrients” before submission.

Abstract:

Authors should briefly justify their study indicating the background – such information should be added (one sentence will be enough).

R:  Following your instruction, we added a sentence to introduce the background of this study.

Number of batches/ samples/ repetitions should be specified.

R: Thanks for your comments.

We have supplemented the description about the replicated determination.

Introduction:

The sentences in the second part of the manuscript seem to be not “combined”. In a paragraph each sentence should “follow” the previous one, while in this part they seem as just random information. It seems that Authors indicate that extracting procedure may influence structure and, as a result, also the biological activity. Authors should more precisely describe it.

In this section, Authors should justify the study, while presenting the history of the problem. In the present form, they present the history of their own past studies. Such justification is in appropriate. If their own studies are the only one that were conducted – it should be described.

R: Thanks for your constructive comments.

We have made some corrections following your suggestions.

In introduction section, we summarized some published papers about the alfalfa polysaccharide (Aspinall and McGrath, 1966; Lamsal et al., 2007; Chen et al., 2015), and also introduced some findings that obtained in our lab.

  In the past several years, we have always devoted on exploring the alfalfa polysaccharide extracted and purified methods, and its bioactivities. We modified the traditional extract procedure and the modified method has obtained the Chinese national invention patent (Patent No. 201610415782.4), and the bioactivities regarding the antioxidant capacity and growth-promoting function to animals have been published (Wang et al., 2019; Zhang et al., 2019)

Materials and Methods:

Number of batches/ samples/ repetitions should be specified.

R: Thanks for your comments.

We have supplemented the description about replicated determination.

Figure 1 – The photographs do not present any information – they should be removed from the scheme, as Authors should briefly present most important data with no illustrations.

R: Thanks for your comments.

According to the reviewer instruction, we removed the photographs and made some revises to the figure.

If there is more than one sample/ repetition analysed (see above – it is not known), there are also following problems: It seems, that Authors did not verify the normality of distribution for the assessed variables. Authors must verify the normality of distribution and specify the test applied for verification. If the distribution is normal, the mean values should be presented (accompanied by SD), but if it is different than normal, the median, accompanied by minimum and maximum values should be presented – it should be specified that distribution is normal if it is. The applied test should be chosen taking into account the observed distribution.

R: Thanks for your insightful comments.

As mentioned above, there were 5 replicates for each sample determination. According to reviewer’s suggestion, we verified the normality of observed data using Shapiro-Wilk test, and all the data were normally distributed. Results were expressed as mean ± SEM with 5 replicated determinations. The significance of differences between means was evaluated by one-way analysis of variance (ANOVA) and Tukey’s multiple comparison test using SAS version 9.0. We revised the description of the statistical method in re-submitted manuscript.

Results:

The whole section should be corrected accordingly taking into account the major problems specified above.

If the distribution is normal, the mean values should be presented (accompanied by SD), but if it is different than normal, the median, accompanied by minimum and maximum values should be presented – it should be specified that distribution is normal if it is. The applied test should be chosen taking into account the observed distribution.

 Figure 6 (a-c) – data should be rather presented as a table to be easier to follow.

R: According to the reviewer’s comment, we carefully went through the results and revised some mistakes to make the manuscript clear.

According to reviewer’s suggestion, we verified the normality of observed data using Shapiro-Wilk test, and all the data were normally distributed. Results were expressed as mean ± SD with 5 replicates. The significant differences between means were evaluated by one-way analysis of variance (ANOVA) and Tukey’s multiple comparison test using SAS version 9.0. We revised the description of the statistical method.

Figure 6 showed the activation of APS to T and B lymphocytes and expression of cytokines. In this section, the proliferation and cytokines expression of splenic T, B cells subjected to APS/LPS-treated under different APS concentration or different incubation time were determined. The finding of proliferation were indicated by OD values, which shown in Figure style were much more clear shown the difference of concentrate and incubation time.

Discussion:

The section should be corrected accordingly and Authors should objectively assess the value of their results taking into account the number of batches/ samples/ repetitions. Authors should extensively discuss the limitations of their study.

R: Thanks for your insightful comments. We carefully checked the discussion and have made substantial corrections. In this section, the innovation of the present research was present and made compare with other similar researches.

Conclusions:

Authors should not reproduce results in this section, but they should present the broader conclusion instead (1-3 sentences will be enough).

R: Thanks for your kind comments.

According to the reviewer’s suggestions, we revised the conclusion statement to make the manuscript clear much more.

Authors Contributions:

Authors should correct this section in order to indicate clearly which Author contributed. E.g. who is “GZ”? do Authors mean GGZ? Authors should be consistent.

R: Thanks for your comments. We carefully checked and revised the problem.

References

Aspinall, G. O., McGrath, D., 1966. The hemicelluloses of lucerne. Journal of the Chemical Society C: Organic, 2133-2139.

Chen, L., Liu, J., Zhang, Y., Dai, B., An, Y., Yu, L. L., 2015. Structural, thermal, and anti-inflammatory properties of a novel pectic polysaccharide from alfalfa (Medicago sativa L.) stem. J. Agr. Food Chem. 63, 3219-3228.

Kanwal, S., Joseph, T. P., Owusu, L., Xiaomeng, R., Meiqi, L., Yi, X., 2018. A Polysaccharide Isolated from Dictyophora indusiata Promotes Recovery from Antibiotic-Driven Intestinal Dysbiosis and Improves Gut Epithelial Barrier Function in a Mouse Model. Nutrients

 10, 1003.

Lamsal, B. P., Koegel, R. G., Gunasekaran, S., 2007. Some physicochemical and functional properties of alfalfa soluble leaf proteins. LWT-Food Sci. Technol. 40, 1520-1526.

Park, J. Y., Shin, M. S., Kim, S. N., Kim, H. Y., Kim, K. H., Shin, K. S., Kang, K. S., 2016. Polysaccharides from Korean Citrus hallabong peels inhibit angiogenesis and breast cancer cell migration. Int. J. Biol. Macromol. 85, 522-529.

Wang, L., Xie, Y., Yang, W., Yang, Z., Jiang, S., Zhang, C., Zhang, G., 2019. Alfalfa polysaccharide prevents HO-induced oxidative damage in MEFs by activating MAPK/Nrf2 signaling pathways and suppressing NF-?B signaling pathways. Sci. Rep.-UK 9, 1782.

Zhang, C. Y., Gan, L. P., Du, M. Y., Shang, Q. H., Xie, Y. H., Zhang, G. G., 2019. Effects of dietary supplementation of alfalfa polysaccharides on growth performance, small intestinal enzyme activities, morphology, and large intestinal selected microbiota of piglets. Livest. Sci. 223, 47-52.

Zheng, Y., Zhang, Q., Liu, X., Ma, L., Lai, F., 2016. Extraction of polysaccharides and its antitumor activity on Magnolia kwangsiensis Figlar & Noot. Carbohyd. Polym. 142, 98-104.

Reviewer 2 Report

The extraction, purification and characterisation of a polysaccharide fraction from M. sativa is described, along with an investigation of its effects on lymphocytes. The polysaccharide has a complex nature, largely made up of galacturonic acid and glucose with smaller amounts of other monosaccharides.

The experimental strategy is on the whole well described, with some problems as listed below. The polysaccharide has no effect on the proliferation or cytokine expression of T-cells and only a slight effect on B cells.

Major problem:

Section 2.3: In the first paragraph of section 2.3, the statement “It can be described according to the following parameters: the number-average molecular weight (Mn) is generally used in kinetics studies and stoichiometric calculations; the weight-average molecular weight (Mw) indicates the tensile strength of the polysaccharide; and the Z-average molecular weight (Mz), also referred to as flexural life, represents the ability of a polymer to repeatedly bend before breaking” is not correct. This is a misunderstanding of some examples given in the (uncited) paper by Oberlerchner et al. Molecules 2015, 20, 10313-10341. The sentence should be deleted and replaced by the simple definitions of Mw, Mn and Mz as given in Oberlerchner et al. equations 1, 2 and 3. Please also cite this paper in the references. It is not good scientific behaviour to use material from another group’s work without acknowledgement.

The purified APS samples were subject to a complex work-up before GPC. Why was this done? Some explanation is needed.

Minor problems:

Tables 1 and 3: the abbreviation ‘nc’ is not defined.

In addition, in Table 3 columns ‘nc’ and ‘nc*min’ values are given to far too many significant figures. At least please round to the nearest whole number.

Page 8, text just above figure 4: the sentence beginning “The intersection of LS and RI curves …” should be deleted. This intersection has no scientific meaning, and is dependent on the relative vertical scale of the graphs.

In Fig. 4, please explain why the RI signal does not come down to baseline after the polysaccharide peak.

Fig. 5: Is ‘Gla’ a misprint for ‘Gal’?

The proposed structure assumes that the fraction prepared by the authors is a single polydisperse polysaccharide with a regular structure. Please explain how the authors know that this is not a mixture of two or more polysaccharides.

Abbreviations: Please use IUPAC-IUB approved abbreviations for uronic acids, GalA and GlcA, throughout the manuscript.

Author Response

Dear editor and reviewers:

I would like to extend our thanks to you for taking the time to carefully review our manuscript (nutrients-486076) titled “Extract Methods, Molecular Characteristics, and Bioactivities of Polysaccharide from Alfalfa (Medicago sativa L.)”. We are grateful for the helpful feedback.

We have carefully read the comments and suggestions, and revised our manuscript line-by-line following those comments and the “Guidelines for Authors” of the Journal. Following reviewers’ comments, all corrected portions are highlighted with “Tracked Changes” pattern in the re-uploaded manuscript. Before submitting, we checked our article according to the ‘instructions to authors’ of the journal, and asked a professional English editor (MDPI English editing) to help us to revise it.

We hope that these revisions have improved the paper such that you now deem it worthy of publication in “Nutrients.

If you have any questions, please do not hesitate to contact us. Once again, thanks for your detailed comments and suggestions to our manuscript.

Sincerely,

Guiguo Zhang, Ph.D.

Department of Nutrition, University of California (UC-Davis),

One Shields Ave, Davis, CA 95616, USA;

Department of Animal Nutrition, Shandong Agricultural University

Taian City, Shandong Province, 271000, China  

Tel : +86-13954859206

Email: zhanggg@sdau.edu.cn;      guzhang@ucdavis.edu

Reviewer 2

Comments and Suggestions for Authors

The extraction, purification and characterisation of a polysaccharide fraction from M. sativa is described, along with an investigation of its effects on lymphocytes. The polysaccharide has a complex nature, largely made up of galacturonic acid and glucose with smaller amounts of other monosaccharides.

The experimental strategy is on the whole well described, with some problems as listed below. The polysaccharide has no effect on the proliferation or cytokine expression of T-cells and only a slight effect on B cells.

R: Thanks for your kind comments.

Major problem:

Section 2.3: In the first paragraph of section 2.3, the statement “It can be described according to the following parameters: the number-average molecular weight (Mn) is generally used in kinetics studies and stoichiometric calculations; the weight-average molecular weight (Mw) indicates the tensile strength of the polysaccharide; and the Z-average molecular weight (Mz), also referred to as flexural life, represents the ability of a polymer to repeatedly bend before breaking” is not correct. This is a misunderstanding of some examples given in the (uncited) paper by Oberlerchner et al. Molecules 2015, 20, 10313-10341. The sentence should be deleted and replaced by the simple definitions of Mw, Mn and Mz as given in Oberlerchner et al. equations 1, 2 and 3. Please also cite this paper in the references. It is not good scientific behaviour to use material from another group’s work without acknowledgement.

R: thanks for your insightful comments!

We have corrected the statement following the Oberlerchner et al. (2015) description, and cited the valuable paper.

The purified APS samples were subject to a complex work-up before GPC. Why was this done? Some explanation is needed.

R: In the determination of macromolecular polysaccharides or starch, the purpose of pretreatment is to make a complete dissolution of starch in some solvent. In present, the 90% Dimethyl sulfoxide (DMSO) is considered as the most commonly used solvents, and adding low molecular weight electrolyte (NaNO3 or LiBr) into DMSO can be used as a cosolvent to improve the solubility of starch. So in this study, we just used this method to treat the APS before determining the molar mass.

Minor problems:

Tables 1 and 3: the abbreviation ‘nc’ is not defined.

R:  “nc” (nano coulomb) is the unit of quantity of electric charge. In chromatographic determination, the unit of quantity of electric charge (nc) is usually used to represent the peak height. We have supplemented the abbreviation.

In addition, in Table 3 columns ‘nc’ and ‘nc*min’ values are given to far too many significant figures. At least please round to the nearest whole number.

R: Under the same retention time, the finding presented the same glycosidic residue in 5 samples determination. The saccharide residue and glycosidic linkage remained unchanged. So there were no SD values for the retention time. 

Page 8, text just above figure 4: the sentence beginning “The intersection of LS and RI curves …” should be deleted. This intersection has no scientific meaning, and is dependent on the relative vertical scale of the graphs.

R: Thanks for your comments. We have deleted the sentence.

In Fig. 4, please explain why the RI signal does not come down to baseline after the polysaccharide peak.

R: RI is the refractive index, and the signal strength is proportional to the concentration of the substance. After 55 minutes, the rising of the refractive index line indicates that the substance was eluted, but the red laser scattering line was very low, indicating that the molecular weight of the eluted substance was very small, which is speculated to be some small molecular impurities, such as metal ions.

Fig. 5: Is ‘Gla’ a misprint for ‘Gal’?

R: Thanks for your insightful comments. We have corrected the error.

The proposed structure assumes that the fraction prepared by the authors is a single polydisperse polysaccharide with a regular structure. Please explain how the authors know that this is not a mixture of two or more polysaccharides.

R: According to the analysis to polysaccharide compositions, the molar ratio of each component, the glycosidic linkage between the saccharide residues (indicating the information of terminal residues and the intrachain residues), and the molar ratio of each residues, can give a primary profile about the polysaccharide molecular, which was corresponding to the molar mass determined in this study. So we preliminary concluded that this is a single polydisperse polysaccharide with a regular structure, rather than a mixture. It needs to be pointed out, however, that further study might be needed to decipher whether the molecular structure is different under different extraction methods.

Abbreviations: Please use IUPAC-IUB approved abbreviations for uronic acids, GalA and GlcA, throughout the manuscript.

R: Thanks for your insightful comments. We have corrected the term in IUPAC-IUB approved abbreviations.

References

Oberlerchner, J. T., Rosenau, T., Potthast, A., 2015. Overview of methods for the direct molar mass determination of cellulose. Molecules (Basel, Switzerland) 20, 10313-10341.

Round 2

Reviewer 1 Report

The manuscript entitled “Extract Methods, Molecular Characteristics, and Bioactivities of Polysaccharide from Alfalfa (Medicago sativa L.)” presents interesting issue, but it requires some important corrections.

Major:

1.       Authors must clearly specify the number of (1) batches (batch is a bunch of product from e.g. one field, or one producer), (2) samples (sample is piece of product taken from one batch) and (3) repetitions (repetition is reproduced measurement of the same sample), as they are not the same. In the present version, Authors present them as exactly the same (e.g. “Five batches samples were replicated”) and reader does not know what exactly was measured

General:

Some parts of the manuscript are shabbily prepared (e.g. lacking spaces, or redundant spaces, leadings, References section, etc.)

Abstract:

Number of batches/ samples/ repetitions should be specified.

Introduction:

The sentences in the second part of the manuscript seem to be not “combined”. In a paragraph each sentence should “follow” the previous one, while in this part they seem as just random information.

Materials and Methods:

Number of batches/ samples/ repetitions should be specified.

Results:

Figure 6 (a-c) – data should be rather presented as a table to be easier to follow.

Discussion:
The section should be corrected accordingly and Authors should objectively assess the value of their results taking into account the number of batches/ samples/ repetitions.

Authors should extensively discuss the limitations of their study.

Author Response

Dear editor and reviewers:

I would like to extend our thanks to you for taking the time to carefully review our manuscript (nutrients-486076) titled “Extract Methods, Molecular Characteristics, and Bioactivities of Polysaccharide from Alfalfa (Medicago sativa L.)”. We are grateful for the helpful feedback.

We have carefully read the comments and suggestions, and revised our manuscript line-by-line following those comments and the “Guidelines for Authors” of the Journal. Following reviewers’ comments, all corrected portions are highlighted with “Tracked Changes” pattern in the re-uploaded manuscript. Before submitting, we checked our article according to the ‘instructions to authors’ of the journal, and asked a professional English editor (MDPI English editing) to help us to revise it.

We hope that these revisions have improved the paper such that you now deem it worthy of publication in “Nutrients.

If you have any questions, please do not hesitate to contact us. Once again, thanks for your detailed comments and suggestions to our manuscript.

Sincerely,

Guiguo Zhang, Ph.D.

Department of Nutrition, University of California (UC-Davis),

One Shields Ave, Davis, CA 95616, USA;

Department of Animal Nutrition, Shandong Agricultural University

Taian City, Shandong Province, 271000, China  

Tel : +86-13954859206

Email: zhanggg@sdau.edu.cn;      guzhang@ucdavis.edu

Comments:

The manuscript entitled “Extract Methods, Molecular Characteristics, and Bioactivities of Polysaccharide from Alfalfa (Medicago sativa L.)” presents interesting issue, but it requires some important corrections.

 Major:

1. Authors must clearly specify the number of (1) batches (batch is a bunch of product from e.g. one field, or one producer), (2) samples (sample is piece of product taken from one batch) and (3) repetitions (repetition is reproduced measurement of the same sample), as they are not the same. In the present version, Authors present them as exactly the same (e.g. “Five batches samples were replicated”) and reader does not know what exactly was measured

R: Thanks for your constructive comments.

Following the reviewer’s comments, we have re-written the section, and gave a detailed description in Material and Methods, and accordingly corrected the statement in abstract and discussion.

General:

Some parts of the manuscript are shabbily prepared (e.g. lacking spaces, or redundant spaces, leadings, References section, etc.)

R: Thanks for your comments.

We have carefully checked the re-uploaded manuscript sentence-by-sentence, and asked MDPI English editor to revise the language and format for us. I hope the meticulously revised manuscript can meet your requirement.  

Abstract:

Number of batches/ samples/ repetitions should be specified.

R: Following the reviewer’s instructions, we have corrected the abstract.

Introduction:

The sentences in the second part of the manuscript seem to be not “combined”. In a paragraph each sentence should “follow” the previous one, while in this part they seem as just random information.

R: Thanks for your insightful comments.

Following the reviewer’s instructions, we made a substantial correction to the introduction, almost re-wrote the section. Adding some content which related to the present study, and deleting some tedious descriptions. We hope the carefully revised manuscript can meet the requirement.

Materials and Methods:

Number of batches/ samples/ repetitions should be specified.

R: Thanks for your comments.

We re-wrote the section according to the reviewer’s instructions.  

Results:

Figure 6 (a-c) – data should be rather presented as a table to be easier to follow.

 R: Thanks for your comments.

We have presented those results in a table following the reviewer’s instructions.

Discussion:

The section should be corrected accordingly and Authors should objectively assess the value of their results taking into account the number of batches/ samples/ repetitions.

Authors should extensively discuss the limitations of their study.

R: Thanks for your comments.

Following the reviewer’s comments, we revised the discussion.

In this section, referred to some published paper in this field, we compared the results in this study and other relevant publications, and analysize the might reasons for the discrepancy in compositions and molecular structures among different studies. In the present study, the different area alfalfa samples have the same polysaccharide compositions and molecular characteristics. Similarly, some literatures found the compositions and molecular properties of APS had no difference between years[1]. However, whether there are some variations for the APS from different growing period still need to be investigated. Because the nutrients such as cellulose, hemicellulose, protein, will be vary with the plant growing. We have discussion the point in the re-submitted manuscript.

Once again, thanks for your insightful comments. Those comments are all valuable and very helpful for revising and improving our paper, as well as the crucial guiding significance to our future researches.

References

1.     Rovkina, K.I.; Krivoshchekov, S.V.; Guryev, A.M.; Yusubov, M.S.; Belousov, M.V. Water-Soluble Polysaccharides of Alfalfa (Medicago sativa (Fabaceae)) of Flora of Krasnoyarsk Krai. Russ. J. Bioorg. Chem. 2018, 44, 854-859.
